

# The predictive value of PRDM2 in solid tumor: a systematic review and meta-analysis

Caroline Tanadi[1], Alfredo Bambang[2], Indra Putra Wendi[2],
Veronika M. Sidharta[3], Linawati Hananta[4] and Anton Sumarpo[2]

[1] Undergraduate Medical Program, School of Medicine and Health Sciences, Atma Jaya Catholic University of Indonesia, Jakarta, Indonesia
[2] Department of Chemistry and Biochemistry, School of Medicine and Health Sciences, Atma Jaya Catholic University of Indonesia, Jakarta, Indonesia
[3] Department of Histology, School of Medicine and Health Sciences, Atma Jaya Catholic University of Indonesia, Jakarta, Indonesia
[4] Department of Pharmacology and Pharmacy, School of Medicine and Health Sciences, Atma Jaya Catholic University of Indonesia, Jakarta, Indonesia

Corresponding author
Anton Sumarpo,
anton.sumarpo@atmajaya.ac.id

## ABSTRACT

**Background.** Many studies have reported the presence of Positive Regulatory/Su(var)3-9, Enhancer-of-zeste and Trithorax Domain 2 (PRDM2) downregulation in cancer. However, its potential as a diagnostic biomarker is still unclear. Hence, a systematic review and meta-analysis were conducted to address this issue.

**Introduction.** As of 2018, cancer has become the second leading cause of death worldwide. Thus, cancer control is exceptionally vital in reducing mortality. One such example is through early diagnosis of cancer using tumor biomarkers. Having a function as a tumor suppressor gene (TSG), *PRDM2* has been linked with carcinogenesis in several solid tumor. This study aims to assess the relationship between *PRDM2* downregulation and solid tumor, its relationship with clinicopathological data, and its potential as a diagnostic biomarker. This study also aims to evaluate the quality of the studies, data reliability and confidence in cumulative evidence.

**Materials & Methods.** A protocol of this study is registered at the International Prospective Register of Systematic Reviews (PROSPERO) with the following registration number: CRD42019132156. PRISMA was used as a guideline to conduct this review. A comprehensive electronic search was performed from inception to June 2019 in Pubmed, Cochrane Library, ProQuest, EBSCO and ScienceDirect. Studies were screened and included studies were identified based on the criteria made. Finally, data synthesis and quality assessment were conducted.

**Results.** There is a significant relationship between *PRDM2* downregulation with solid tumor (RR 4.29, 95% CI [2.58–7.13], $P < 0.00001$). The overall sensitivity and specificity of *PRDM2* downregulation in solid tumors is 84% (95% CI [39–98%]) and 86% (95% CI [71–94%]), respectively. There is a low risk of bias for the studies used. TSA results suggested the presence of marked imprecision. The overall quality of evidence for this study is very low.

**Discussion.** We present the first meta-analysis that investigated the potential of *PRDM2* downregulation as a diagnostic biomarker in solid tumor. In line with previous studies,

our results demonstrated that *PRDM2* downregulation occurs in solid tumor. A major source of limitation in this study is the small number of studies.

**Conclusions**. Our review suggested that *PRDM2* is downregulated in solid tumor. The relationship between *PRDM2* downregulation and clinicopathological data is still inconclusive. Although the sensitivity and specificity of *PRDM2* downregulation are imprecise, its high values, in addition to the evidence that suggested *PRDM2* downregulation in solid tumor, hinted that it might still have a potential to be used as a diagnostic biomarker. In order to further strengthen these findings, more research regarding *PRDM2* in solid tumors are encouraged.

# INTRODUCTION

Cancer has long been considered a catastrophic public health problem due to its high mortality rates. With an estimate of 9.6 million death, cancer has now become the second leading cause of death worldwide. Moreover, the incidence of cancer is also growing at an alarming rate due to the exponential increase of the aging population and changes in the distribution of cancer risk factors. It was estimated that the incidence of cancer would rise to 18.1 million new cases in 2018. To summarise, one in six women and one in five men will develop cancer, while one in 10 women and one in eight men are dying as a result of cancer (*Bray et al., 2018*; *World Health Organization, 2018*).

Thus, cancer control is extremely vital in reducing mortality. One example of cancer control is early diagnosis of cancer. This could be achieved through the use of tumor biomarkers. However, despite the potential of biomarkers for early detection of cancer, its implementation in the clinical setting is still lacking (*Goossens et al., 2015*; *Schiffman, Fisher & Gibbs, 2015*; *World Health Organization, 2017*). This could be attributed to weak clinical performances, such as low sensitivity, low specificity or low predictive values (*Diamandis, 2012*). Hence, further research to identify novel biomarkers should be performed.

Positive Regulatory/Su(var)3-9, Enhancer-of-zeste and Trithorax Domain 2 (PRDM2) is a tumor suppressor gene (TSG) that regulates protein expression through the methylation of lysine 9 in histone H3. Hence, *PRDM2* also belongs to the nuclear histone/protein methyltransferase superfamily. Its gene products are also involved in DNA-binding and transcription factor binding-activities, implicating its role in carcinogenesis (*Sorrentino et al., 2018*; *Zhang et al., 2015*). Studies have also reported *PRDM2* downregulation in cancers that exhibit high incidence and mortality, such as bladder cancer, breast cancer, cervical cancer, colorectal cancer, endometrial cancer, esophageal squamous cell carcinoma, gastric carcinoma, hepatocellular carcinoma, lung cancer, pancreatic cancer, prostate cancer, T-cell prolymphocytic leukemia and thyroid carcinoma (*Cheng, Gao & Lou, 2010*; *Cui et al., 2016*; *Johansson et al., 2018*; *Lal et al., 2006*; *Michalak & Visvader, 2016*; *Oshimo et al., 2004*; *Pandzic et al., 2017*; *Rossi et al., 2009*; *Sakurada et al., 2001*; *Tan et al., 2014*; *Wu et al.,*
*2016*; *Yang et al., 2017*; *Zhang et al., 2016*). Furthermore, in a meta-analysis that found a total of 22 genes methylated in hepatocellular carcinoma, *PRDM2* was one of the genes with the most significant result and is on par with the well-known *APC* and *p16* (*Zhang et al., 2016*). Hence, *PRDM2* might play an important role in malignancies. However, the potential of *PRDM2* as a diagnostic biomarker is still unclear.

Therefore, we performed a systematic review and meta-analysis that investigated *PRDM2* expression level in solid tumor, as well as its potential as a diagnostic biomarker. If there is sufficient data, we will also investigate if there is any correlation between *PRDM2* expression level with clinicopathological data.

## MATERIAL AND METHODS

### Study registration and methodology

A protocol of this study is registered at the International Prospective Register of Systematic Reviews (PROSPERO) with the following registration number: CRD42019132156 (https://www.crd.york.ac.uk/prospero/display_record.php?RecordID=132156) (*National Institute for Health Research, 2019*). Preferred Reporting Items for Systematic Reviews and Meta-analyses (PRISMA) flow diagram was used as a guideline to conduct our systematic review and meta-analysis (*Moher et al., 2009*).

### Search strategy and study selection

A comprehensive electronic search was done in PubMed, Cochrane Library, ProQuest, EBSCO and ScienceDirect from inception to July 2019 using the following search terms: (PRDM2 OR RIZ OR RIZ1 OR RIZ2 OR KMT8 OR KMT8A OR MTB-ZF OR HUMHOXY1) AND (Cancer OR Cancers OR Malignant OR Malignancy OR Malignancies OR Neoplasm OR Neoplasms OR Neoplasia OR Neoplasias OR Tumor OR Tumors OR Tumour OR Tumours). The search was performed by two independent reviewers (Alfredo Bambang and Indra Putra Wendi). Any differences were solved through a discussion with a third reviewer (Anton Sumarpo).

All of the search outputs were exported into the EndNote software. Duplicates were removed, and screening was performed based on the title and abstract of the study. Probable or included studies were identified and assessed for eligibility according to the criteria above. Finally, included studies were identified, and data extraction was performed.

A study is included if it meets the following criteria: (1) The study used human subjects; (2) The study investigated the relationship between *PRDM2* expression level and solid tumor through the use of gene expression analysis; (3) The study used histopathological examination as a comparator; (4) The study is a clinical trial or cross-sectional study. A study is excluded if: (1) The study does not have a control group (people without cancer or non-cancer specimens); (2) The study did not use an appropriate or did not state the gene expression analysis method used; (3) The expression level of *PRDM2* in the study is not clearly stated or unquantifiable; (4) The study is a review, case series, conference abstracts, *in vitro* or *in vivo* study. (5) The study is not written in English.

## Data extraction

The included studies were then analyzed further and the following informations are extracted: First author, publication year, country of origin, age, gender, race, type of cancer, cancer differentiation state, stage of cancer, type of control, number of cases and controls, gene expression analysis method, *PRDM2* expression level and conclusion of the study. In the case of missing data, the authors will be contacted via email to request access to those missing data.

## Data synthesis and statistical analysis

Sensitivity and specificity of *PDRM2* were assessed in order to elucidate the potential of *PRDM2* expression level as a diagnostic biomarker in solid tumor. Sensitivity and specificity are said to be significant if > 50%. Risk ratio (RR) with a 95% confidence interval (CI) was used to determine the relationship between *PRDM2* expression level and risk of cancer, as well as the relationship between *PRDM2* expression level and clinicopathological data. If heterogeneity is present, Random Effects Model (REM) will be used. However, if heterogeneity is absent, Fixed Effects Model (FEM) will be used instead.

Cochrane's Q test (chi-squared test) and Higgins $I^2$ statistics were used to assess for the presence of heterogeneity statistically. Heterogeneity is said to be present if $P < 0.10$ or $I^2 > 75\%$ (*Higgins & Green, 2011*; *Higgins et al., 2003*). To assess for the presence of heterogeneity visually, a forest plot will be generated. Meta-regression and subgroup analysis will be conducted when there are at least 10 studies used in the meta-analysis (*Baker et al., 2009*). The possible causes of heterogeneity are: Age, gender, ethnicity, country of origin, type of cancer, cancer differentiation state, stage of cancer and genotyping method.

Funnel plot and Deek's test will be used to assess publication bias when the number of included studies is at least 10. If the funnel plot is asymmetric, publication bias is present. If the $P$-value for Deek's test is $< 0.10$, there is funnel plot asymmetry (*Deeks, Macaskill & Irwig, 2005*). If publication bias is found, the trim and fill method will be used to correct this bias (*Duval & Tweedie, 2000*).

Furthermore, sensitivity analysis was performed to elucidate the effect and stability of a single study on the pooled estimates by deleting one study at a time. Additionally, sensitivity analysis was also conducted to compare the pooled estimates using odds ratio (OR) and RR, as well as using REM and FEM. All statistical analyses were generated using RevMan 5.3 and STATA 12.0.

## Quality assessment and data reliability

In order to claim that the meta-analysis conducted has been conclusive, the required information size has to be achieved. Thus, a trial sequential analysis (TSA) was performed using TSA software in order to determine the required information size (*Wetterslev, Jakobsen & Gluud, 2017*). Quality of evidence will be assessed using Quality Assessment of Diagnostic Accuracy Studies-2 (QUADAS-2) which consists of the following key domains: patient selection, index test, reference standard, as well as flow and timing (*Whiting et al., 2011*).

## Confidence in cumulative evidence

Grading of Recommendations, Assessment, Development, and Evaluations (GRADE) was used to evaluate the confidence in cumulative evidence. Overall certainty of evidence can be written as high, moderate, low or very low (*Schünemann et al., 2013*).

## RESULTS

### Search results

Using variants of the keywords "PRDM2" and "cancer", we performed a search from inception to July 2019 in PubMed, Cochrane Library, ProQuest, EBSCO and ScienceDirect. After duplicate removal, a total of 3,928 records was obtained. Titles and abstracts were screened and 58 potential studies were identified. Out of these 58 studies, 52 were excluded due to the studies being unable to meet the inclusion criteria (ineligible), *in vitro* and/or *in vivo*, used unsuitable methods, written in non-English, or is a review. The remaining six studies (*Akahira et al., 2007*; *Dong et al., 2012*; *Ge et al., 2015*; *Geli et al., 2005*; *Jiang et al., 1999*; *Tan et al., 2018*) were included in the systematic review while only five studies (*Akahira et al., 2007*; *Dong et al., 2012*; *Geli et al., 2005*; *Jiang et al., 1999*; *Tan et al., 2018*) were included in the meta-analysis. This is because *Ge et al. (2015)* did not mention the number of samples and controls that expressed *PRDM2* downregulation in renal cell carcinoma. Thus, only five studies were included in the meta-analysis. The Preferred Reporting Items for Systematic Reviews and Meta-analyses (PRISMA) flow diagram for this study is shown in Fig. 1.

The studies that were eligible for systematic review were published from 1999 to 2015. There were a total of 314 samples of solid tumors and 225 controls obtained from patients in China (*Dong et al., 2012*; *Ge et al., 2015*; *Tan et al., 2018*), Japan (*Akahira et al., 2007*), Sweden (*Geli et al., 2005*) and United States of America (*Jiang et al., 1999*). All of these six studies are cross-sectional studies. The solid tumors included in this study are ovarian cancer (*Akahira et al., 2007*), esophageal squamous cell carcinoma (*Dong et al., 2012*), renal cell carcinoma (RCC) (*Ge et al., 2015*), pheochromocytoma (*Geli et al., 2005*), abdominal paraganglioma (*Geli et al., 2005*), hepatoma (*Jiang et al., 1999*), lung squamous cell carcinoma (LSCC) (*Tan et al., 2018*) and lung adenocarcinoma (LAC) (*Tan et al., 2018*). Out of these six studies, one used immunohistochemistry (IHC) only (*Akahira et al., 2007*), three used reverse transcription-polymerase chain reaction (RT-PCR) only (*Ge et al., 2015*; *Geli et al., 2005*; *Jiang et al., 1999*) and two used both IHC and RT-PCR (*Dong et al., 2012*; *Tan et al., 2018*). A summary of the main characteristics of the included studies for systematic review and meta-analysis is presented in Tables 1 and 2, respectively.

### Systematic review results

All six studies concluded that *PRDM2* gene expression is significantly decreased in solid tumor compared to control, with the *P*-value ranging from $< 0.05$ to $< 0.001$ using CI 95%. *Akahira et al. (2007)* stated that there was a significant correlation between *PRDM2* downregulation with cancer grade ($P < 0.0345$) and stage ($P < 0.0153$) in ovarian cancer. On the other hand, *Ge et al. (2015)* stated otherwise, concluding that there was no significant relationship between RCC with tumor progression ($P = 0.19$). A study by *Geli et al. (2005)*
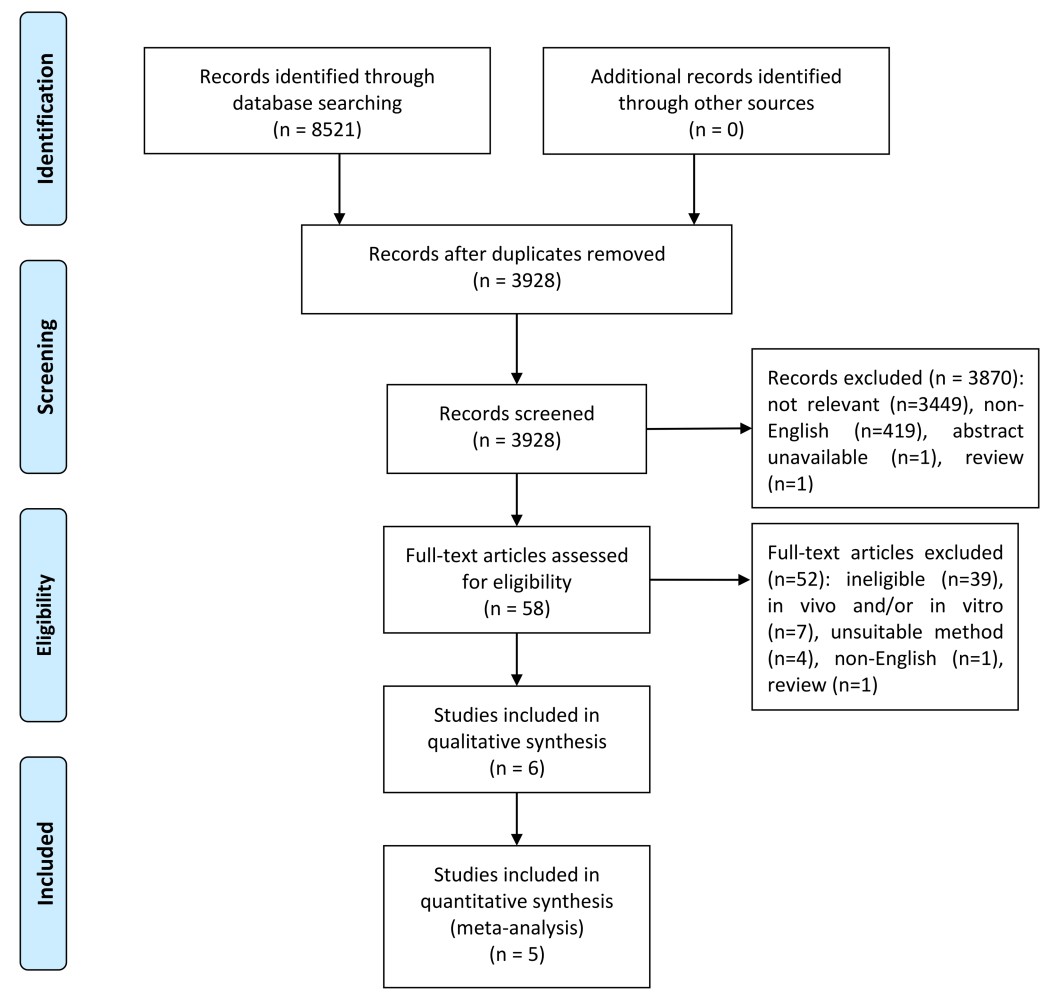

**Figure 1** **PRISMA flow diagram for selection of included studies.**

reported that decreased *PRDM2* gene expression was not correlated significantly with gender and tumor size, but was found to be weakly correlated with younger age (Spearman rank-order correlations; $R = 0.4$). Other clinicopathological data were either absent or not investigated in the studies. Hence, the role of *PRDM2* downregulation in cancer grade, stage, gender, age and other clinicopathological data is still unclear. Due to the lack of sufficient clinicopathological data, only *PRDM2* gene expression and its sensitivity and specificity were further analysed in the meta-analysis.

## Meta-analysis results

Five studies were included in this meta-analysis to further investigate the relationship between *PRDM2* downregulation with solid tumor (*Akahira et al., 2007*; *Dong et al., 2012*; *Geli et al., 2005*; *Jiang et al., 1999*; *Tan et al., 2018*). The pooled analysis suggested that *PRDM2* gene expression is decreased in solid tumor (RR 4.29, 95% CI [2.58–7.13], $P < 0.00001$; Fig. 2). Based on this pooled analysis, three sensitivity analyses were conducted

Tanadi et al. (2020), *PeerJ*, DOI 10.7717/peerj.8826

**Table 1   Study characteristics of studies included in systematic review.**

| Study | Country | Age | Gender | Race | Method | No. of sample | No. of control | Cancer | | | | | | PRDM2 expression | P value |
|---|---|---|---|---|---|---|---|---|---|---|---|---|---|---|---|
| | | | | | | | | Type | | Stage | | Differentiation | | | |
| | | | | | | | | | | I + II | III + IV | Well/ Moderate | Poor | | |
| *Akahira et al. (2007)* | Japan | <50 (n = 42/67) >=50 (n = 68/97) | ND | Asian | IHC | 164 | 6[a] | Ovarian cancer | | 69 | 95 | 107[h] | 36[h] | Decreased[i] | <0.05 |
| *Dong et al. (2012)* | China | ND | ND | Asian | RT-PCR IHC | 40 | 40[b] | Esophageal squamous cell carcinoma | | ND | ND | ND | ND | Decreased[i] | <0.05[j] |
| *Ge et al. (2015)* | China | ND | ND | Asian | qRT-PCR | 20 | 20[c] | Renal cell carcinoma | | ND | ND | ND | ND | Decreased[i] | <0.001[k] |
| *Geli et al. (2005)* | Sweden | ND | 7 M 4 F | Caucasian | qRT-PCR | 11 | 6[d] | Pheochromocytoma (n = 4) Abdominal paraganglioma (n = 7) | | ND | ND | ND | ND | Decreased[i] | <0.001[l] |
| *Jiang et al. (1999)* | United States of America | ND | ND | Caucasian | RT-PCR | 4 | 3[e] | Hepatoma | | ND | ND | ND | ND | Decreased[i] | ND |
| *Tan et al. (2018)* | China | <60 (n = 30) >=60 (n = 45) | 56 M 19 F | Asian | RT-PCR IHC | 75 | 150[f] | LSCC (n = 52) LAC (n = 23) | | 63[g] | 12[g] | 46[g] | 29[g] | Decreased[i] | <0.05[m] |

**Notes.**

[a] Normal ovaries.

[b] Adjacent non-cancerous tissue.

[c] Adjacent non-malignant renal tissue.

[d] Normal adrenal cells.

[e] Normal liver tissue.

[f] Tumor adjacent tissue and distant lung tissue.

[g] Classification based on International Association for the Study of Lung Cancer 2009.

[h] Classification based on universal grading system for ovarian epithelian cancer.

[i] PRDM2 expression level is decreased when compared to control.

[j] Chi-square test; $X^2 = 12.00$.

[k] Median fold difference = 0.08 (interquartile range 0.03–0.50).

[l] Wilcoxon matched pair test.

[m] Student's $t$-test or one-way analysis of variance, followed by Newman-Keuls test.

F, Female; IHC, Immunohistochemistry; LAC, Lung adenocarcinoma; LSCC, Lung squamous cell carcinoma; M, Male; ND, Not determined; PRDM2, Positive Regulatory/Su(var)3-9, Enhancer-of-zeste and Trithorax Domain 2; qRT-PCR, Quantitative reverse transcription-polymerase chain reaction; RT-PCR, Reverse transcription-polymerase chain reaction.

**Table 2  Study characteristics of studies included in meta-analysis.**

| Study | Method | No. of sample | No. of control | Cancer type | TP | FP | FN | TN |
|---|---|---|---|---|---|---|---|---|
| *Akahira et al. (2007)* | IHC | 164 | 6 | Ovarian cancer | 110 | 0 | 54 | 6 |
| *Dong et al. (2012)* | IHC | 12 | 12 | Esophageal squamous cell carcinoma | 12 | 4 | 0 | 8 |
| *Geli et al. (2005)* | qRT-PCR | 11 | 6 | Pheochromocytoma ($n = 4$) Abdominal paraganglioma ($n = 7$) | 9 | 0 | 2 | 6 |
| *Jiang et al. (1999)* | qRT-PCR | 4 | 3 | Hepatoma | 4 | 1 | 0 | 2 |
| *Tan et al. (2018)* | IHC | 75 | 150 | LSCC ($n = 52$) LAC ($n = 23$) | 22 | 10 | 53 | 140 |

**Notes.**

FN, False negative; FP, False positive; IHC, Immunohistochemistry; LAC, Lung adenocarcinoma; LSCC, Lung squamous cell carcinoma; qRT-PCR, Quantitative reverse transcription-polymerase chain reaction; TN, True negative; TP, True positive.

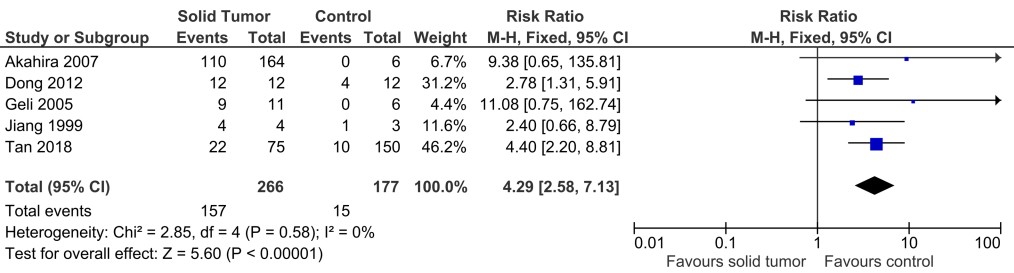

**Figure 2  Forest plot of *PRDM2* downregulation in solid tumors and control.** Studies with notable weights are *Tan et al. (2018)* (46.2%) and *Dong et al. (2012)* (31.2%). The results from this forest plot demonstrated that *PRDM2* downregulation occurs more often in solid tumor when compared to control (RR 4.29, 95% CI [2.58–7.13], $P < 0.00001$). There was no significant heterogeneity in this analysis ($X^2 = 2.85$, $I^2 = 0\%$). The horizontal line represents 95% CI. The blue box is the result of each individual study. The black diamond at the bottom of the plot is the pooled analysis of all studies. CI, Confidence interval. df, Degree of freedom. $I^2$, Test of heterogeneity. M-H, Mantel-Haenszel.

to evaluate the stability of our findings: with and without the deletion of *Jiang et al. (1999)* (Fig. 3), RR or OR (Fig. 4), and FEM or REM (Fig. 5). All three sensitivity analyses did not have meaningful differences, proving that our results are stable.

The sensitivity and specificity of *PRDM2* downregulation in solid tumor were also assessed in order to investigate its potential as a diagnostic biomarker. A split forest plot displaying the sensitivity and specificity of the included studies is shown in Fig. 6. As demonstrated in the summary receiver operating characteristic (SROC) curve (Fig. 7), the summary sensitivity and specificity of decreased *PRDM2* gene expression in solid tumor is 84% (95% CI [39–98]%) and 86% (95% CI [71–94]%), respectively. This result is in favor of *PRDM2* downregulation as a potential diagnostic biomarker. However, the confidence interval for *PRDM2* downregulation is wide, suggesting that there is marked imprecision. This was later confirmed on TSA (Fig. 8). In Fig. 8, the line representing the cumulative $Z$-curve failed to cross the significance boundary and did not reach the required number of studies which is 7743. Therefore, it can be concluded that the usage of *PRDM2* downregulation as a diagnostic biomarker in solid tumor is still inconclusive.

**A**

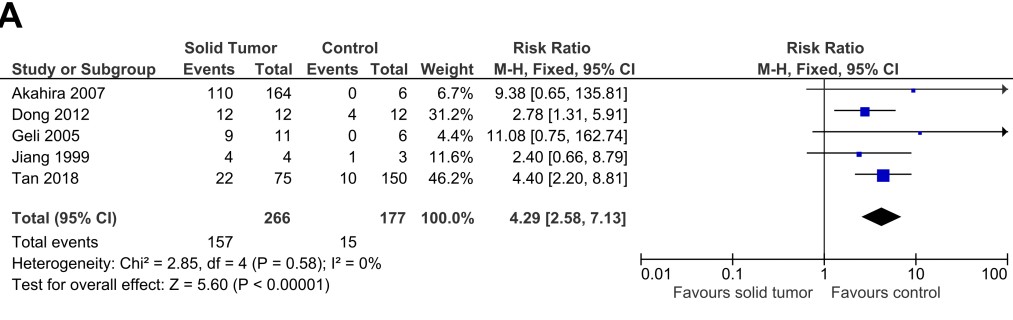

**B**

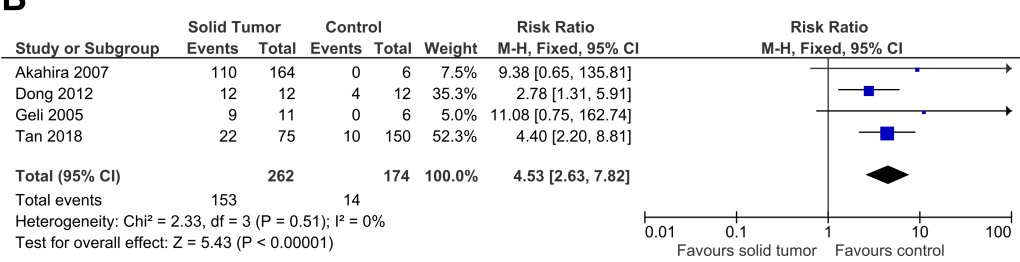

**Figure 3 Sensitivity analysis to compare the use of all studies with deletion of a study.** The deleted study, *Jiang et al. (1999)*, is a study that has the most questionable results based on the risk of bias assessment. There was a significant result for both analyses: (A) Without deletion of *Jiang et al. (1999)*: RR 4.29, 95% CI [2.58–7.13], $P < 0.00001$, $X^2 = 2.85$, $P < 0.58$, $I^2 = 0\%$; (B) With deletion of *Jiang et al. (1999)*: RR 4.53, 95% CI [2.63–7.82], $P < 0.00001$, $X^2 = 2.33$, $P < 0.51$, $I^2 = 0\%$. The deletion of *Jiang et al. (1999)* increased RR by 1.1 times higher with the 95% CI 1.2 times wider. The deletion of study also slightly lowered heterogeneity. This sensitivity analysis proved that the results were stable. The horizontal line represents 95% CI. The blue box is the result of each individual study. CI, Confidence interval. df, Degree of freedom. $I^2$, Test of heterogeneity. M-H, Mantel-Haenszel.

## Quality assessment of included studies

The quality of the included studied was evaluated using the QUADAS-2 tool, and a summary of the results can be viewed in Table 3. As shown in Table 3, in the index test domain there are four studies (*Ge et al., 2015*; *Geli et al., 2005*; *Jiang et al., 1999*; *Tan et al., 2018*) having an unclear risk of bias. These four studies did not directly state whether the index test (gene expression analysis) was interpreted independently from the reference standard (histopathological examination). Thus, we decided that unclear was most fit as the risk of bias. One of the studies, *Jiang et al. (1999)* also had missing information on how the patients were recruited, leading to an unclear risk of bias for one other domain. In general, the quality of the included studies was robust, ensuring the reliability of our systematic review and meta-analysis.

## Confidence in cumulative evidence

By assessing five domains, including the risk of bias (by using the results from QUADAS-2 risk of bias assessment), indirectness, inconsistency, imprecision (by using the results from TSA) and risk of publication bias, a GRADE evidence profile was constructed as shown in Table 4. To be noted, all of the included studies used diagnostic accuracy test as their design, whereby all of the samples and controls will undergo both the index test and reference

**A**

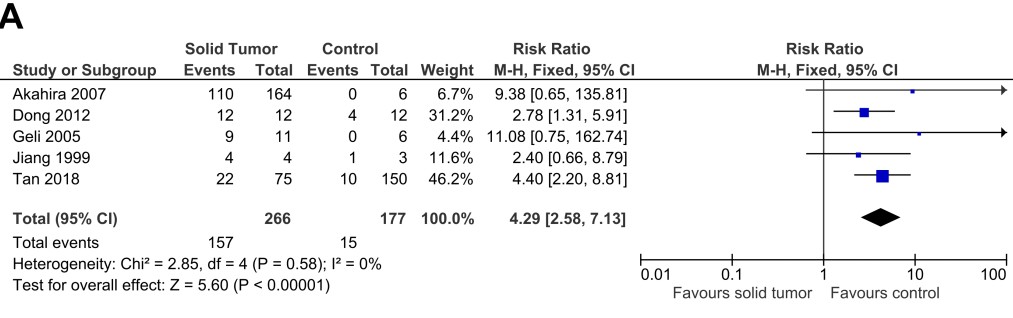

**B**

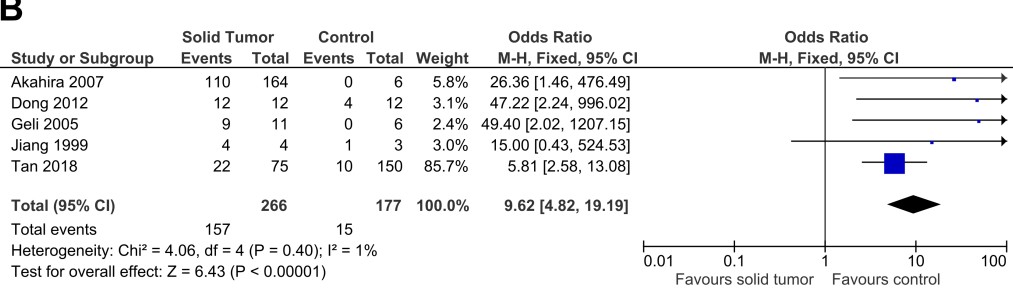

**Figure 4** **Sensitivity analysis to compare the use of Risk Ratio (RR) with Odds Ratio (OR).** There was a significant result for both analyses: (A) RR: RR 4.29, 95% CI [2.58–7.13], $P < 0.00001$; (B) OR: OR 9.62, 95% CI [4.82–19.19], $P < 0.00001$. The use of OR gave a result two times higher with the 95% CI three times wider when compared to RR. RR had a slightly lower heterogeneity when compared to OR (RR: $X^2 = 2.58$, $P < 0.58$, $I^2 = 0\%$; OR: $X^2 = 4.06$, $P < 0.40$, $I^2 = 1\%$). This sensitivity analysis proved that the results were stable. The horizontal line represents 95% CI. The blue box is the result of each individual study. The black diamond at the bottom of the plot is the pooled analysis of all studies. CI, Confidence interval. df, Degree of freedom. $I^2$, Test of heterogeneity. M-H, Mantel-Haenszel.

standard. Ideally, diagnostic studies should randomize which of the samples and controls will undergo the index test only and which will undergo the reference standard only. Hence, this made the design susceptible to indirectness. In addition, most of the included studies have wide confidence interval and inconclusive TSA results. Thus, serious was placed in the imprecision domain. As for publication bias, since the number of included studies is < 10, publication bias could not be evaluated. Unfortunately, this does not entirely rule out the possibility of publication bias being present in our study, and thus we decided to downgrade the quality of evidence further. Overall, we have very low confidence in the pooled estimates obtained for our meta-analysis.

# DISCUSSION

In this study, we have successfully generated the first meta-analysis that investigated the potential of *PRDM2* downregulation as a diagnostic biomarker in solid tumor. Compared to previous primary studies on *PRDM2* thus far, we investigated the significance of *PRDM2* with solid tumor on the level of a review. This includes the evaluation of quality assessment, data reliability and confidence in cumulative evidence, proving that our study was more comprehensive.

**A**

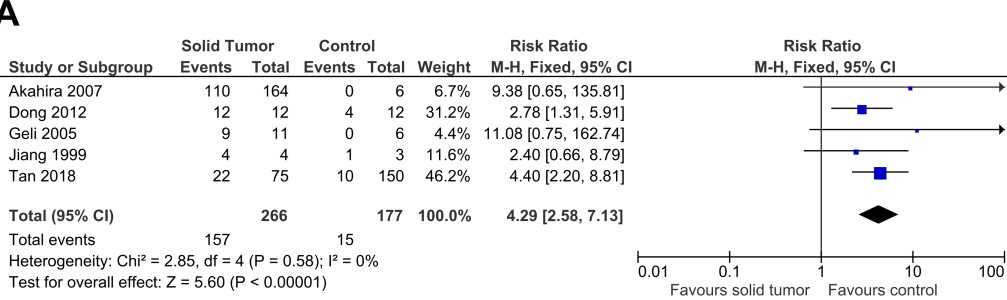

**B**

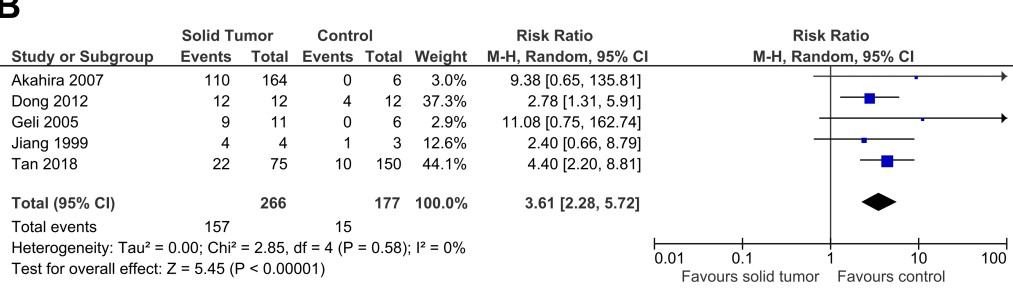

**Figure 5** **Sensitivity analysis to compare the use of Fixed Effects Model (FEM) with Random Effects Model (REM).** There was a significant result for both analyses: (A) FEM: RR 4.29, 95% CI [2.58–7.13], $P < 0.00001$; (B) REM: RR 3.61, 95% CI [2.28–5.72], $P < 0.00001$. FEM increased RR by 1.2 times higher with 95% CI 1.3 times wider. This sensitivity analysis proved that the results were stable. The horizontal line represents 95% CI. The blue box is the result of each individual study. The black diamond at the bottom of the plot is the pooled analysis of all studies. CI, Confidence interval. df, Degree of freedom. $I^2$, Test of heterogeneity. M-H, Mantel-Haenszel.

| Study | TP | FP | FN | TN | Sensitivity (95% CI) | Specificity (95% CI) | Sensitivity (95% CI) | Specificity (95% CI) |
|---|---|---|---|---|---|---|---|---|
| Akahira 2007 | 110 | 0 | 54 | 6 | 0.67 [0.59, 0.74] | 1.00 [0.54, 1.00] | | |
| Dong 2012 | 12 | 4 | 0 | 8 | 1.00 [0.74, 1.00] | 0.67 [0.35, 0.90] | | |
| Geli 2005 | 9 | 0 | 2 | 6 | 0.82 [0.48, 0.98] | 1.00 [0.54, 1.00] | | |
| Jiang 1999 | 4 | 1 | 0 | 2 | 1.00 [0.40, 1.00] | 0.67 [0.09, 0.99] | | |
| Tan 2018 | 22 | 10 | 53 | 140 | 0.29 [0.19, 0.41] | 0.93 [0.88, 0.97] | | |

**Figure 6** **Forest plot for sensitivity and specificity of decreased *PRDM2* gene expression in solid tumor.** Studies that have high sensitivities include *Dong et al. (2012)* (Sensitivity 1.00, 95% CI [0.74–1.00]) and *Jiang et al. (1999)* (Sensitivity 1.00, 95% CI [0.40–1.00]). Studies that have high specificities are *Akahira et al. (2007)* (Specificity 1.00, 95% CI [0.54–1.00]) and *Geli et al. (2005)* (Specificity 1.00, 95% CI [0.54–1.00]). The horizontal line represents 95% CI. The blue box is the result of each individual study. CI, Confidence interval. FN, False negative. FP, False positive. TN, True negative. TP, True positive.

Meta-regression, funnel plot and Deek's test were not performed due to the small number of studies obtained. Due to the inability to confirm the presence of publication bias, we also could not perform trim and fill method. Since our results indicated that there was no heterogeneity in the studies used, a subgroup analysis was not required.

In line with previous studies, our results demonstrated that *PRDM2* downregulation occurs in ovarian cancer, esophageal squamous cell carcinoma, hepatoma and lung cancer. According to *Sorrentino et al. (2018)*, *PRDM2* downregulation has also been reported

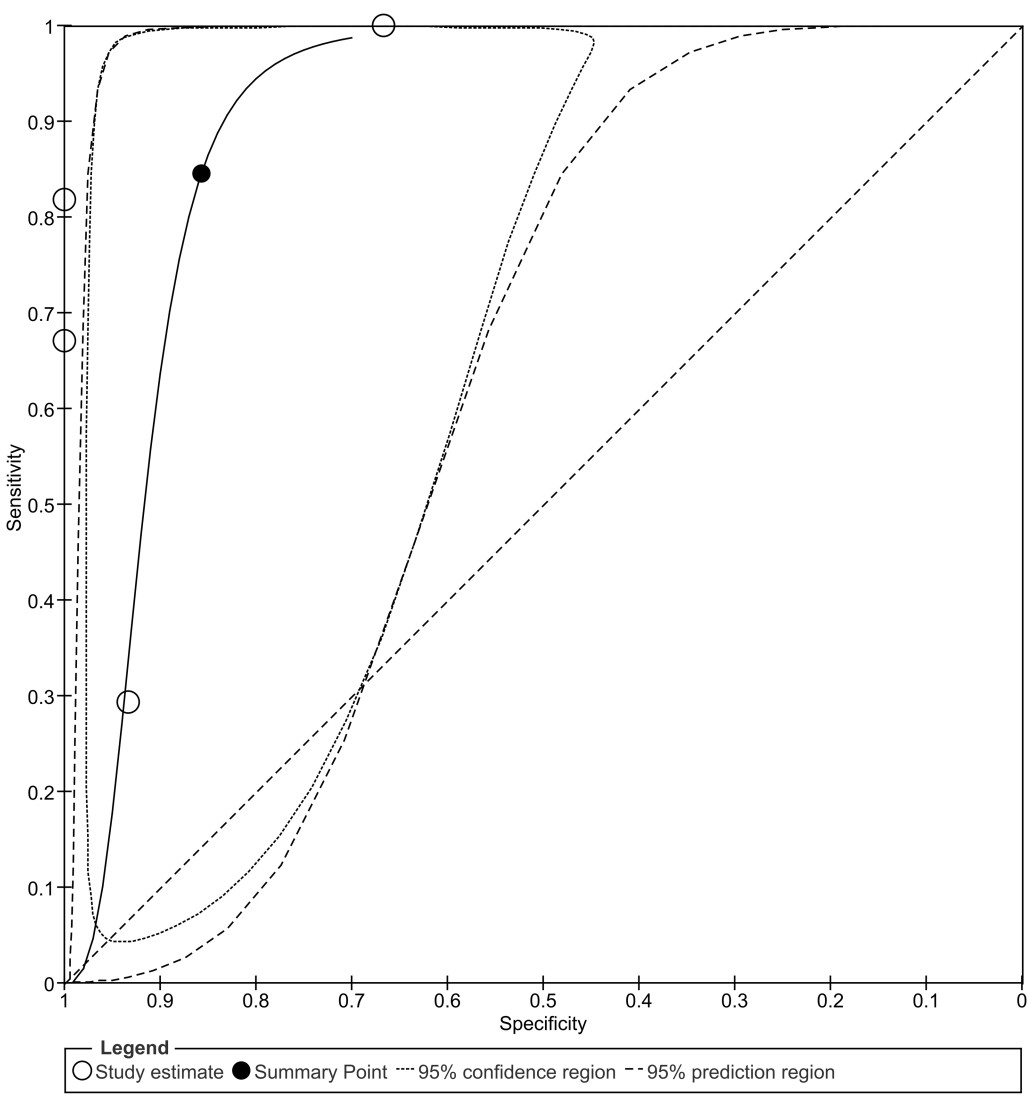

**Figure 7  Summary receiver operating characteristic (SROC) curve of decreased *PRDM2* gene expression in solid tumor.** The overall sensitivity and specificity is 84% (95% CI [39–98]%) and 86% (95% CI [71–94]%), respectively. The calculation of these results can be viewed at Fig. S1. The black circle (summary estimate) represents the summary estimate of sensitivity and specificity. The dotted lines around the summary point represents the 95% confidence region. The dashed lines represent the 95% prediction region (the region within which we are 95% certain that the results of a new study will lie).

in neuroblastoma, breast cancers, melanoma, parathyroid adenoma and Merkel cell carcinoma. However, our included studies did not investigate those solid tumors. Another notable difference is the inconclusive results linking *PRDM2* downregulation with cancer stage and grade even though *PRDM2* downregulation has been associated with cancer progression (*Sun et al., 2011*). A possible explanation for these inconsistencies might be due to the fact that our study only accepted human studies, and thus limited the possibility of encountering such studies. Interestingly, all of the individual studies did not have a

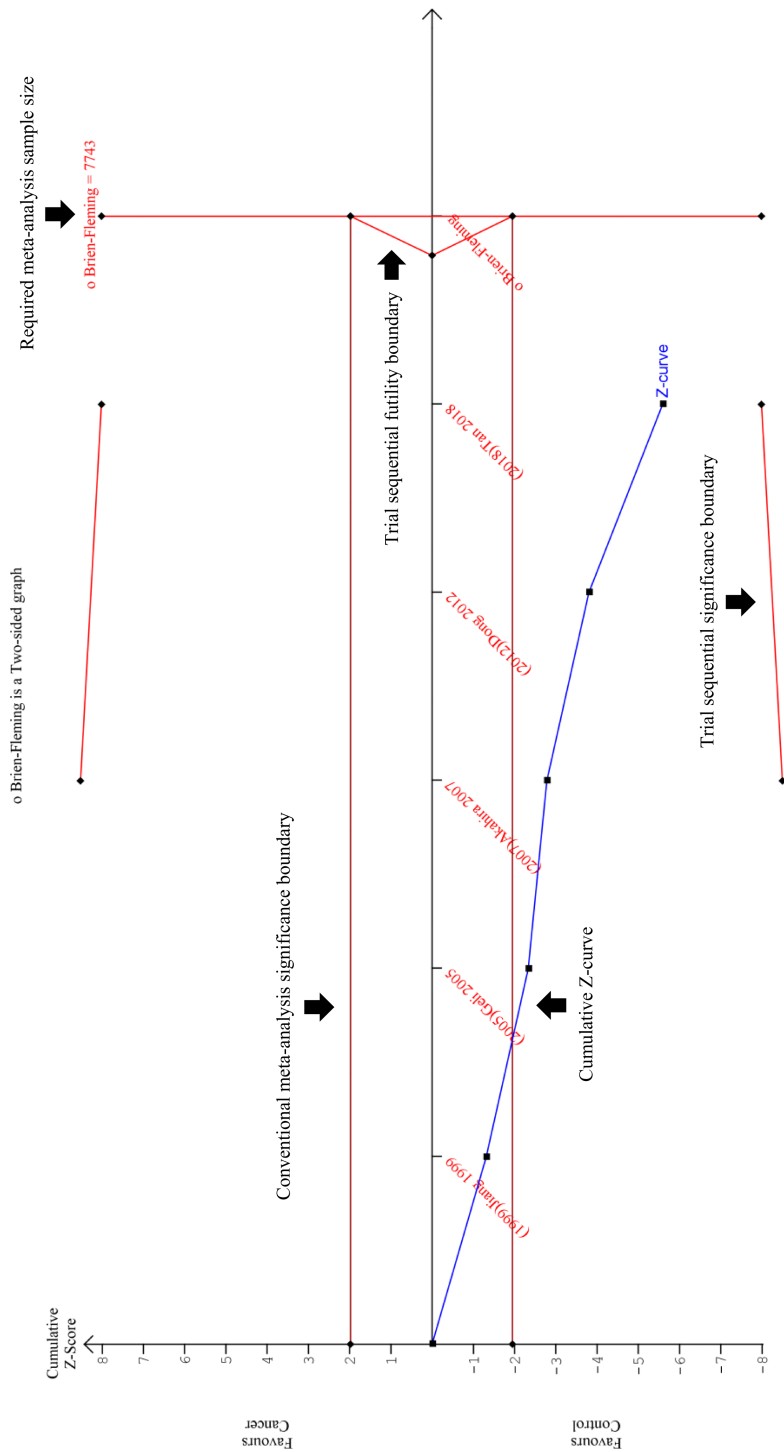

**Figure 8 Trial Sequential Analysis (TSA) results of the meta-analysis.** The cumulative $Z$-curve (blue line) crossed the conventional meta-analysis significance boundary (horizontal red lines at $Z = +1{,}96$ and $Z = -1{,}96$), confirming that type I error was avoided. However, the cumulative $Z$-curve has not crossed the trial sequential significance boundary (diagonal red line at the top and bottom of the plot), suggesting (continued on next page...)

**Figure 8 (...continued)**
that type II error might have not been avoided. Furthermore, the cumulative $Z$-score has also failed to reach the vertical red line on the right, indicating that this review has not reached the required sample size which is 7743. It is interesting to note that the cumulative $Z$-curve did not cross the trial sequential futility boundary (triangular red line on the right), implying that the addition of new samples could potentially improve the TSA results. In conclusion, this TSA analysis proved that this meta-analysis still requires more samples in order ensure that type II error was avoided. This is a magnified version of the TSA. The TSA results on a standard scale can be viewed at Fig. S2.

**Table 3  Quality Assessment of Diagnostic Accuracy Studies-2 (QUADAS-2) risk of bias assessment.**

| Study | Quality Assessment of Diagnostic Accuracy Studies-2 (QUADAS-2) | | | | | | |
|---|---|---|---|---|---|---|---|
| | Risk of bias | | | | Applicability concerns | | |
| | Patient selection | Index test | Reference standard | Flow and timing | Patient selection | Index test | Reference standard |
| *Akahira et al. (2007)* | Low | Low | Low | Low | Low | Low | Low |
| *Dong et al. (2012)* | Low | Low | Low | Low | Low | Low | Low |
| *Ge et al. (2015)* | Low | Unclear | Low | Low | Low | Low | Low |
| *Geli et al. (2005)* | Low | Unclear | Low | Low | Low | Low | Low |
| *Jiang et al. (1999)* | Unclear | Unclear | Low | Low | Low | Low | Low |
| *Tan et al. (2018)* | Low | Unclear | Low | Low | Low | Low | Low |

standardised baseline to define *PRDM2* downregulation. Although this could lead to possible heterogeneity, our study demonstrated otherwise.

Following these findings, an important question to address is whether *PRDM2* downregulation could be used as a diagnostic biomarker in solid tumor. As described above, the high sensitivity and specificity of *PRDM2* downregulation suggested its potential as a diagnostic biomarker. However, these values have wide confidence intervals and inconclusive TSA results, implying there was marked imprecision (*Chai-Adisaksopha, Thorlund & Iorio, 2016*; *Tan & Tan, 2010*). Thus, the use of *PRDM2* downregulation as a diagnostic biomarker is still inconclusive. This imprecision might be due to the small number of sample and controls used in the individual studies or low variability in the subjects used (*Carlson & Morrison, 2009*). In addition, there was also a vast difference between the sample and control size, whereby the sample size is much larger. We believe that this was because some of the studies did not obtain their sample and control from the same subject. This made acquirement of control samples, such as normal ovaries or normal adrenal cells, much more difficult when compared to pathological samples that are readily retrieved for examination. Although our present study could not fully prove the potential of *PRDM2* downregulation as a diagnostic biomarker due to its imprecision, it is important to highlight that these results can potentially improve with the addition of new studies. This has been proven by our TSA results whereby the line representing the cumulative $Z$-curve did not cross the futility boundary.

Another issue that should be addressed in the future is whether the quality of our evidence is satisfying enough. The quality of evidence is judged based on five domains: risk of bias, indirectness, inconsistency, imprecision and publication bias. It should be

Tanadi et al. (2020), *PeerJ*, DOI 10.7717/peerj.8826

**Table 4 Grading of Recommendations, Assessment, Development, and Evaluations (GRADE) evidence profile for the studies included in the meta-analysis.**

| Outcome | No. of studies | Design | Grading of Recommendations, Assessment, Development, and Evaluations (GRADE) | | | | | |
| --- | --- | --- | --- | --- | --- | --- | --- | --- |
| | | | Risk of bias | Indirectness | Inconsistency | Imprecision | Publication bias | Quality of evidence |
| True positives (patients/samples with solid tumor) | 5 studies (443 patients/samples) | Cross-sectional studies | Not serious | Serious[a] | Not serious | Serious[b] | ND[c] | ⊕○○○ Very low |
| False negatives (patients/samples incorrectly classified as not having solid tumor) | 5 studies (443 patients/samples) | Cross-sectional studies | Not serious | Serious[a] | Not serious | Serious[b] | ND[c] | ⊕○○○ Very low |
| True negatives (patients/samples without solid tumor) | 5 studies (443 patients/samples) | Cross-sectional studies | Not serious | Serious[a] | Not serious | Serious[b] | ND[c] | ⊕○○○ Very low |
| False positives (patients/samples incorrectly classified as having solid tumor) | 5 studies (443 patients/samples) | Cross-sectional studies | Not serious | Serious[a] | Not serious | Serious[b] | ND[c] | ⊕○○○ Very low |

**Notes.**

[a]All samples undergo both index test and reference standard, introducing indirectness into the studies.

[b]Most of the individual studies have a wide confidence interval and inconclusive TSA results.

[c]Publication bias could not be evaluated as the number of studies is <10.

ND, Not determined; QUADAS-2, Quality Assessment of Diagnostic Accuracy Studies-2; TSA, Trial sequential analysis.

noted that all of the studies used in this review are diagnostic accuracy studies which are considered a proxy to randomised-controlled trials. Hence, indirectness is present, and this could lead to overestimation of sensitivity and specificity, resulting in the downgrading of the quality of evidence (*Schmidt & Factor, 2013*). As discussed before, imprecision is present, and publication bias could not be assessed, leading to further downgrading. Together, these three domains led to the downgrading of the quality of evidence from high to very low. Although there is very low confidence for our results, it is important to highlight once again that these results can improve if new studies are added.

Limitations of our study are the lack of RCTs as part of our included studies which made it difficult to evaluate the internal validity of our results (*Carlson & Morrison, 2009*). As mentioned before, our study also lacks clinicopathological data in order to assess the potential of *PRDM2* further. Interestingly, none of the included studies investigated *PRDM2* gene expression in the same type of solid tumor. Hence, we were unable to evaluate in which type of solid tumor is *PRDM2* downregulation most suitable to be used as a biomarker. Furthermore, there was no standardised baseline among studies. Another limitation of this study involves the issue of only using studies written in English, leading to the possibility of language bias. Most of the individual studies have a wide confidence interval and inconclusive TSA results, indicating there is insufficient knowledge about the effect and that further research should be done. Based on the points above, it can be concluded that a major source of limitation is due to the small number of studies.

## CONCLUSIONS

In conclusion, our review suggested that *PRDM2* gene expression is decreased or downregulated in solid tumor. Due to insufficient data, we are unable to determine the relationship between *PRDM2* downregulation and clinicopathological data. Although the sensitivity and specificity of *PRDM2* downregulation are imprecise, its high values, in addition to the evidence that suggested *PRDM2* downregulation in solid tumor, hinted that it might still have a potential to be used as a diagnostic biomarker. Furthermore, its imprecision could potentially be solved through the addition of new studies. Thus, we suggest more research to be conducted, especially those with RCT as their design, to fully elucidate the potential of *PRDM2* downregulation in solid tumor. More study is urgently needed to determine a standardised baseline for *PRDM2* downregulation level. We would also recommend more research regarding the relationship between *PRDM2* gene expression with clinicopathological data to further evaluate the potential of *PRDM2* gene expression in solid tumor. Finally, once there is sufficient data available, we suggest a new systematic review and meta-analysis to be done in order to renew the findings of our study.

**Abbreviations**

| | |
|---|---|
| **CI** | Confidence interval |
| **Df** | Degree of freedom |
| **F** | Female |
| **FN** | False negative |

| | |
|---|---|
| **FP** | False positive |
| **GRADE** | Grading of recommendations, assessment, development, and evaluations |
| **IHC** | Immunohistochemistry |
| **LAC** | Lung adenocarcinoma |
| **LSCC** | Lung squamous cell carcinoma |
| **M** | Male |
| **M-H** | Mantel-Haenszel |
| **ND** | Not determined |
| **OR** | Odds ratio |
| **PRDM2** | Positive regulatory/su(var)3-9, enhancer-of-zeste and trithorax domain 2 |
| **PRISMA** | Preferred reporting items for systematic reviews and meta-analyses |
| **PROSPERO** | International prospective register of systematic reviews |
| **qRT-PCR** | Quantitative reverse transcription-polymerase chain reaction |
| **QUADAS-2** | Quality assessment of diagnostic accuracy studies-2 |
| **RCC** | Renal cell carcinoma |
| **REM** | Random effects model |
| **RR** | Risk ratio |
| **RT-PCR** | Reverse transcription-polymerase chain reaction |
| **SROC** | Summary receiver operating characteristic |
| **TN** | True negative |
| **TP** | True positive |
| **TSA** | Trial sequential analysis |
| **TSG** | Tumor suppressor gene |

## ACKNOWLEDGEMENTS

The authors would like to acknowledge School of Medicine and Health Sciences, Atma Jaya Catholic University of Indonesia for all the support for this research project.

### Funding

The authors received no funding for this work.

### Competing Interests

The authors declare there are no competing interests.

### Author Contributions

- Caroline Tanadi conceived and designed the experiments, analyzed the data, prepared figures and/or tables, authored or reviewed drafts of the paper, and approved the final draft.
- Alfredo Bambang performed the experiments, prepared figures and/or tables, authored or reviewed drafts of the paper, and approved the final draft.
- Indra Putra Wendi performed the experiments, analyzed the data, prepared figures and/or tables, authored or reviewed drafts of the paper, and approved the final draft.

- Veronika M. Sidharta and Linawati Hananta conceived and designed the experiments, authored or reviewed drafts of the paper, and approved the final draft.
- Anton Sumarpo conceived and designed the experiments, performed the experiments, authored or reviewed drafts of the paper, and approved the final draft.

## Data Availability

The raw data of the study selection (such as which ones are screened based on title/abstract only or based on full text) and the QUADAS-2 scoring of included studies are available in the Supplemental Files.

## Supplemental Information

Supplemental information for this article can be found online at http://dx.doi.org/10.7717/peerj.8826#supplemental-information.

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
