# Peer review of "The predictive value of PRDM2 in solid tumor: a systematic review and meta-analysis"

_PeerJ, doi:10.7717/peerj.8826_

## Round 0.1 · original submission · Minor Revisions

Please address issues raised by two reviewers and revise manuscript accordingly.

Reviewer 1 ·

Basic reporting

The authors undertake a comprehensive analysis to identify if down regulation of PRDM2, a member of the nuclear histone/protein methyltransferase, can serve as a biomarker in early cancer diagnosis.

Experimental design

The study is well thought through and critical of itself. I applaud the authors meticulous and tedious work. I'm in favor of publication as it insights need for more experimental studies on PRDM2 down regulation in solid tumors.

Validity of the findings

the authors should explain in the results how the 3870 records were excluded and they should make conclusions once there is sufficient data available otherwise it appears to be phrased as if authors doubt their own conclusions in their present study.

Annotated reviews are not available for download in order to protect the identity of reviewers who chose to remain anonymous.

Reviewer 2 ·

Basic reporting

The predictive value of PRDM2 in solid tumor: a systematic review and meta-analysis by Tanadi et. al., is a nicely written review presenting a meta-analyses and investigating the potential of PRDM2 downregulation as a diagnostic biomarker in solid tumor. There are no grammatical errors and the writing is crisp. The figures and tables are of high quality. The review addresses important questions and findings pertaining to relationship between PRDM2 and solid tumors and I think it will be of general interest to a reader in cancer research. I have highlighted few important points, which, I believe, need to be addressed before the manuscript is deemed fit to be published in PeerJ.

Experimental design

Within aims and scope of the journal.

Validity of the findings

Meets standards. I have listed a few points in the next section which I believe need to be addressed before it makes it into PeerJ.

Additional comments

Line 61-63: Can the authors give a new recent reference if it is available. Or is the 2018 reference the latest?

Line 66-68: Can the authors comment or add few lines on why the implementation in clinical setting is lacking? It would be of general interest to the reader. For example, highlight the bottlenecks that prevents biomarker discovery translating into clinical setting.

General question pertaining to the Introduction: What is rationale behind selection of PRDM2 for this study over other candidates. Also, cancer is a very broad term. Which specific types of cancer are associated with PRDM2? Or is it a universal biomarker for all cancers?

Line 105-107: “Each step of the process was performed by two independent reviewers (AB and IPW), and any differences were solved through a discussion” – This line seems confusing and I think it’s better to omit this line from the manuscript because the search criteria has been elaborated in the first and third paragraphs of the sub section.

Line124-126, Line 154-155: Similar to the previous comment, the author contributions can be listed at the end of the manuscript. Listing them inside the actual text is not a preferred way.

Line173: Could the authors comment on when they mean by “studies being irrelevant”?

Line 181-184: Why is there so vast difference between the sample size and the control size?

Line 274-277: The authors state that PRDM2 downregulation occurs in solid tumors. What do the authors mean by saying they could not find relationship between PRDM2 and several solid tumors such as neuroblastoma, breast cancers, melanoma, parathyroid adenoma and Merkel cell carcinoma? Is the downregulation? Or no expression of PRDM2 at all? Also, it would important to specify which type of solid tumors show downregulated PRDM2.

Line 286-297: Have the authors tried to analyze whether PRDM2 downregulation can be used as potential diagnostic biomarker in particular type of solid tumor i.e what happens when you don’t group all the solid tumors, what if you analyze solid tumors (for example breast, lung etc.) independently?

Line 315-317: Are there no studies that have used RT-PCR (for example), to quantify the levels of PRDM2 in normal vs solid tumors?

Annotated reviews are not available for download in order to protect the identity of reviewers who chose to remain anonymous.

---

## Round 0.2 · accepted · Accept

In my view, all critiques were addressed and the manuscript was amended accordingly. Therefore, I am pleased to accept this work for publication in PeerJ.